# Neurotransmitter Alterations in Prediabetes and Type 2 Diabetes Mellitus: A Narrative Review

**DOI:** 10.3390/ijms26167847

**Published:** 2025-08-14

**Authors:** Roxana-Viorela Ahrițculesei, Lidia Boldeanu, Anda Lorena Dijmărescu, Mohamed-Zakaria Assani, Mihail Virgil Boldeanu, Isabela Siloși, Cristin Constantin Vere

**Affiliations:** 1Doctoral School, University of Medicine and Pharmacy of Craiova, 200349 Craiova, Romania; roxana.blendea@gmail.com; 2Department of Microbiology, Faculty of Medicine, University of Medicine and Pharmacy of Craiova, 200349 Craiova, Romania; lidia.boldeanu@umfcv.ro; 3Department of Obstetrics and Gynecology, Faculty of Medicine, University of Medicine and Pharmacy of Craiova, 200349 Craiova, Romania; 4Department of Immunology, Faculty of Medicine, University of Medicine and Pharmacy of Craiova, 200349 Craiova, Romania; mihail.boldeanu@umfcv.ro (M.V.B.); isabela_silosi@yahoo.com (I.S.); 5Department of Gastroenterology, University of Medicine and Pharmacy of Craiova, 200349 Craiova, Romania; vere_cristin@yahoo.com

**Keywords:** neurotransmitter, prediabetes, type 2 diabetes mellitus, dopamine, serotonin, norepinephrine, gamma-aminobutyric acid, glutamate

## Abstract

Prediabetes and early type 2 diabetes mellitus (T2D) are increasingly recognized as states of both metabolic and neurochemical dysregulation. This narrative review synthesizes emerging evidence of alterations in key neurotransmitter systems—dopamine, serotonin, norepinephrine, gamma-aminobutyric acid, and glutamate—in individuals with prediabetes and diabetes. Beyond peripheral insulin resistance and β-cell dysfunction, disturbances in the central nervous system, especially related to neurotransmitter signaling, may play a role in disease onset and progression. Neuroimaging studies reveal early imbalances in excitatory and inhibitory neurotransmitters, while biochemical and histological findings demonstrate altered receptor expression in both the brain and pancreatic islets. These changes affect metabolic control and are implicated in mood, cognition, and feeding behavior. We investigate the mechanistic links between neurotransmitter dysfunction and glucose metabolism, including the roles of brain insulin resistance, inflammation, mitochondrial stress, and gut–brain axis signaling. Finally, we discuss therapeutic strategies that target neurochemical pathways and highlight the need for longitudinal, sex-aware, and multi-omics studies to refine early interventions. Understanding the neurobiological roots of early T2D could revolutionize risk assessment and open doors for new neuro-metabolic treatments.

## 1. Introduction

Type 2 diabetes mellitus (T2D) and its precursor, prediabetes, are rapidly growing public health concerns. Globally, more than 470 million adults have impaired glucose regulation—either impaired fasting glucose, impaired glucose tolerance, or both—placing them at heightened risk for developing T2D within five years if no intervention occurs [1,2,3]. Despite the well-established roles of peripheral insulin resistance and β-cell dysfunction in the progression from prediabetes to diabetes, emerging research highlights the importance of central nervous system (CNS) regulation in the early disease stages [4,5,6].

The CNS–metabolism interface has become an increasing research focus over the past decade. Central insulin signaling plays a pivotal role in regulating energy intake, reward-associated feeding behaviors, and peripheral glucose homeostasis. Dysregulation of brain insulin sensitivity—referred to as cerebral insulin resistance—can precipitate maladaptive alterations in feeding behavior and glucose regulation, often preceding the onset of overt hyperglycemia. These findings underscore that molecular perturbations within the CNS may serve as antecedents or causal factors in the development of peripheral metabolic dysregulation during prediabetic states [7,8,9].

At the heart of these CNS mechanisms are pivotal neurotransmitter systems—dopamine, serotonin, norepinephrine, GABA, and glutamate—that modulate feeding behavior, stress responses, and insulin sensitivity. Dopaminergic pathways are crucial for regulating motivational drives related to food intake. Recent experimental and clinical data demonstrate that insulin resistance directly modulates central dopaminergic circuits, resulting in attenuated reward processing and an increased propensity for hyperphagia [10,11]. Intriguingly, dopamine D_2_ receptors in adipose tissue have been upregulated in individuals with prediabetes or new-onset T2D—indicating both central and peripheral dopamine alterations in early disease [12,13].

Serotonin (5-HT) plays a pivotal role in regulating appetite, mood, and glycemic homeostasis, with perturbations evident early in the progression of metabolic disease, even prior to significant hyperglycemia. Concurrently, norepinephrine, predominantly via sympathetic nervous system pathways, modulates hepatic glucose output and insulin sensitivity; emerging evidence suggests that dysregulation of noradrenergic signaling constitutes an early metabolic derangement [14,15,16,17].

Major neurotransmitters, such as gamma-aminobutyric acid (GABA) and glutamate, function ubiquitously within the CNS and play critical roles in pancreatic islet physiology. GABA, synthesized by glutamate decarboxylase (GAD) in both neuronal cells and β-cells, facilitates β-cell viability, suppresses α-cell-derived glucagon secretion, and enhances islet orchestration. Concurrently, glutamate, released from α-cells under hypoglycemic conditions, mediates its effects through ionotropic and metabotropic receptors to modulate insulin and glucagon secretion. These dual roles at central and peripheral sites exemplify an integrated neurotransmitter network which is imperative for maintaining metabolic homeostasis [1,18,19,20,21].

Obesity-related insulin resistance is a key factor in the development of prediabetes and T2D, with increasing evidence pointing to central neuroendocrine dysfunctions in this process [22]. Leptin, a hormone produced by fat cells, acts as a crucial signaling molecule to the hypothalamus, connecting fat accumulation to metabolic control. Normally, when leptin binds in the hypothalamus, it reduces food intake and increases energy expenditure, while also enhancing insulin sensitivity by encouraging glucose uptake and fatty acid oxidation in the liver and muscles [23,24]. Disruption of central leptin signaling significantly impacts glucose metabolism and insulin responsiveness. Typically, leptin’s action in the hypothalamus activates neurocircuitry—such as through autonomic outputs—suppresses hepatic glucose production and boosts insulin sensitivity in peripheral tissues [25]. When this central leptin signal is compromised, these regulatory pathways weaken: hepatic gluconeogenesis becomes less inhibited, and peripheral tissues develop increased insulin resistance, leading to higher blood glucose levels [22,25]. Notably, persistent high leptin levels in obesity can itself cause neuronal insulin resistance by impairing hypothalamic insulin signaling, worsening glucose imbalance [26]. Central leptin resistance results in a condition where appetite is unchecked and glucose regulation is impaired—factors that heighten the risk of prediabetic insulin resistance and hasten the progression to overt T2D [26].

Proton magnetic resonance spectroscopy (1H-MRS) studies show in vivo evidence of early neurotransmitter imbalance in prediabetes. One study found that cerebral GABA-to-glutamate ratios in brainstem areas relate to glycemic control, even in people with mild hyperglycemia. These results suggest that CNS neurotransmitter changes can be identified before the full onset of T2D, highlighting their potential as early mechanistic biomarkers [27,28,29].

Additionally, systematic reviews from 2022 to 2023 highlight that neurotransmitter receptor dynamics in islets and neural circuits are significantly altered during early T2D. These studies record abnormal activity of dopaminergic, glutamatergic, and GABAergic receptors in the brain and the endocrine pancreas [20,30,31].

This body of evidence collectively highlights an evolving paradigm: changes in neurotransmitter systems are not merely a consequence of metabolic dysregulation, but may actively contribute to the initial development of prediabetes and new-onset T2D. Understanding the dynamic changes in CNS and pancreatic islet neurotransmitter environments during early disease stages could help identify new biomarkers and therapeutic targets.

This review thoroughly examines and synthesizes recent findings concerning the roles and alterations of pivotal neurotransmitter systems—namely dopamine, serotonin, NE, GABA, and glutamate—in prediabetic and diabetic states. It critically analyzes central and peripheral neurochemical modifications, their mechanistic contributions to metabolic dysregulation, and their implications for early diagnosis and therapeutic strategies, thereby elucidating the neurobiological substrates underlying initial diabetic pathology. Figure 1 delineates the neurochemical imbalance characteristic of early-stage diabetes.

## 2. Neurotransmitter Systems Involved in Metabolic Regulation

Neurotransmitters—key chemical mediators operating within the CNS and peripheral tissues—are essential for regulating energy balance, blood sugar levels, appetite, and insulin sensitivity. Emerging evidence highlights the complex connection between neurotransmitter signaling pathways and metabolic control networks. Specifically, brain insulin resistance (BIR) disrupts dopaminergic and serotonergic neurotransmission involved in reward processing, which promotes hyperphagia and worsens metabolic dysregulation [11,32,33].

Concurrently, excitatory (glutamate) and inhibitory (GABA) neurotransmitters are implicated in hypothalamic regulation of glucose metabolism. Studies reveal that in individuals with mild hyperglycemia or early T2D, both GABA and glutamate levels in brain regions such as the medial prefrontal cortex become aberrant. For instance, elevated GABA concentrations in the medial prefrontal cortex (mPFC) have been associated with insulin resistance and cognitive impairment in T2D patients. Similarly, glutamate levels correlate with glycemic status, suggesting neurotransmitter imbalance even before the onset of overt diabetes [24,25,26,27,28,29,30,31,32,33,34,35,36].

These modifications indicate that neurotransmitter dysregulation is an early event in prediabetes and new-onset T2D, possibly occurring before and contributing to peripheral metabolic dysfunction rather than just being a result. Understanding the mechanisms behind these neurochemical changes is crucial for discovering new biomarkers and developing targeted treatments at the neuro-metabolic interface.

### 2.1. Dopamine

Dopamine (DA) plays a central role in reward, motivation, and feeding by regulating the mesocorticolimbic and nigrostriatal pathways, which control incentive-driven behaviors and energy intake [11,37,38]. Insulin signaling in the brain is closely linked to dopaminergic function: central insulin resistance impairs dopamine turnover, leading to decreased reward processing, increased anxiety, and depressive-like behaviors in both rodents and humans. These changes may trigger compensatory overeating, contributing to early metabolic imbalance [39].

In peripheral tissues, dopamine modulates insulin secretion and adipocyte function: β-cells co-secrete DA along with insulin, and D2 receptor (D2R) activation inhibits insulin release via heteromeric D1/D2 receptor complexes [11,37,39,40,41]. Notably, D2Rs in subcutaneous adipose tissue are upregulated in individuals with impaired fasting glucose and T2D, correlating with higher HbA1c and insulin resistance—even independent of obesity—suggesting an early compensatory or pathophysiological role [12].

Genetic or pharmacological disruption of D2 signaling in β-cells causes glucose intolerance and impaired insulin response in animal models. Conversely, the targeted modulation of dopaminergic tone with D2 agonists (e.g., bromocriptine, cabergoline, sulpiride) has shown promise in improving glucose homeostasis: bromocriptine remodels adipose tissue metabolism, sulpiride mitigates hyperglycemia in obesity, and cabergoline has even reversed insulin dependence in a human case of autoimmune diabetes [12,13,38,40,42].

#### Central Dopamine Alongside Peripheral Dopamine Receptors and Insulin Sensitivity

A hyperinsulinemic-euglycemic clamp study in healthy adults showed that endogenous dopamine at ventral striatal D2/3 receptors correlated strongly with whole-body insulin sensitivity (r ≈ 0.84) and inversely with plasma insulin; acute dopamine depletion via α-methyl-para-tyrosine reduced the glucose infusion rate and raised insulin levels without altering glycaemia. This may provide causal evidence that impaired central dopaminergic signaling contributes to insulin resistance [43].

Gene expression and protein levels of the D2 receptor in human subcutaneous adipose tissue were significantly higher in individuals with impaired fasting glucose and T2D. D2 receptor expression showed a positive correlation with homeostatic model assessment for insulin resistance (HOMA-IR), fasting insulin, and HbA1c, but a negative correlation with insulin sensitivity. Although dopamine and D2 agonists did not stimulate adipocyte glucose uptake ex vivo, they did suppress catecholamine-induced lipolysis. These human studies collectively demonstrate that dopaminergic dysregulation occurs both centrally and peripherally and is associated with insulin resistance [12].

Emerging evidence suggests that dopamine dysregulation in central reward pathways, pancreatic islets, and adipose tissue acts as a mechanistic link connecting early neurobiological changes to the metabolic problems seen in prediabetes and incident T2D. This highlights the potential of targeting dopaminergic signaling pathways for early therapeutic interventions [11,44,45].

### 2.2. Serotonin (5-HT)

Serotonin (5-hydroxytryptamine, 5-HT) is synthesized peripherally—primarily by enterochromaffin cells in the gut via tryptophan hydroxylase 1—and centrally by raphe nuclei in the brainstem via Tph2. Approximately 90% of total body 5-HT is peripheral, with only ~10% being central [46,47,48]. This dichotomy underlies distinct roles in metabolic regulation.

#### 2.2.1. Skeletal Muscle Signaling and β-Cell Autocrine Signaling

In differentiated L6 skeletal muscle cells, physiologic 5-HT concentrations increased deoxyglucose uptake, promoted GLUT4 translocation, and enhanced glycogen storage. These effects were amplified by insulin and were abolished when protein serotonylation was inhibited. Proteomic analysis revealed serotonylation of the small GTPase Rab4, implicating Rab4 as the convergence point between insulin and serotonin signaling [49].

During pregnancy, tryptophan-hydroxylase-1 in β-cells is upregulated, leading to high intra-islet serotonin. Serotonin acts via the ionotropic 5-HT3 receptor to depolarize β-cells, lower the glucose threshold, and dramatically augment glucose-stimulated insulin secretion. Pregnant mice lacking the Htr3a gene show impaired glucose tolerance despite normal β-cell mass, underscoring the importance of serotonergic signaling for adaptive insulin secretion [50].

#### 2.2.2. Peripheral Actions and Central Actions

Elevated gut-derived 5-HT has been linked to obesity and dysglycemia: obese individuals show increased intestinal 5-HT secretion, which correlates with poor glycemic control [48,51,52]. In turn, peripheral 5-HT promotes hepatic gluconeogenesis, stimulates adipose tissue lipolysis, and affects fat browning—actions that collectively contribute to insulin resistance and hyperglycemia [48,53]. Notably, inhibiting or genetically deleting gut-derived 5-HT synthesis in mice improves glucose tolerance, and changes in gut microbiota that lower peripheral 5-HT are associated with improved glycemic outcomes [54,55].

Furthermore, serotonin acts directly within pancreatic islets; it enhances insulin secretion through the serotonylation of small GTPase proteins in β-cells and inhibits glucagon release via 5-HT_1_f receptors on α-cells [46,48,56]. These dual functions indicate that peripheral 5-HT impacts early islet dysfunction in prediabetes.

Within the CNS, 5-HT modulates appetite, mood, and glucose regulation, especially through 5-HT_2_C receptors in the hypothalamus and nucleus tractus solitarius [46,47,48,52]. Activation of central 5-HT_2_C receptors reduces feeding and controls autonomic outputs; genetic or pharmacologic downregulation increases the risk of hyperphagia and weight gain, which is common in prediabetes [57,58,59,60]. These serotonergic pathways are connected with insulin signaling: insulin resistance impairs central 5-HT synthesis and release, further disrupting satiety and glucose homeostasis [46,61,62,63].

#### 2.2.3. Therapeutic Implications

Selective serotonin reuptake inhibitors (SSRIs), such as sertraline, have been shown to directly boost β-cell insulin secretion and increase β-cell mass in experimental systems [64]. While SSRIs are commonly used for depression, their effects on glucose metabolism are complex—showing both benefits for islet function and potential risks of weight gain—highlighting the therapeutic potential and challenges of targeting the serotonergic system in early T2D [47,48,65]. For example, fluoxetine (Prozac) is an SSRI that can reduce food intake in certain situations; it is notably approved for treating bulimia nervosa and has been demonstrated to decrease binge eating frequency and cause modest weight loss at high doses [66]. The synthesis highlights the diverse roles of serotonin across various physiological systems—including the gastrointestinal tract, liver, adipose tissue, pancreas, and central nervous system—in regulating energy balance and blood sugar levels. Disruption of these interconnected central and peripheral serotonergic pathways seems to be a key factor in the development of prediabetes and early T2D, making them promising targets for therapy [48,62,67,68].

### 2.3. Norepinephrine (NE)

Norepinephrine (NE), the main neurotransmitter of the SNS, has both central effects—mainly through the locus coeruleus—and peripheral actions. It coordinates various physiological responses, including stress adaptation, arousal, cardiovascular regulation, and metabolic processes. In metabolic pathways, NE promotes hepatic gluconeogenesis and glycogenolysis, aids lipolysis in fat cells, influences skeletal muscle glucose uptake, and stimulates glucagon secretion, collectively contributing to hyperglycemia during stress responses [69,70,71,72,73,74].

A hyperinsulinemic-euglycemic clamp study showed that norepinephrine infusion at standard pressor doses decreased the glucose infusion rate from 11.2 ± 3.7 to 9.0 ± 2.6 mg·kg^−1^·min^−1^, without altering steady-state insulin or C-peptide levels. This suggests that heightened sympathetic activity can cause insulin resistance independently of β-cell function [72].

In cultured brown adipocytes, norepinephrine stimulated glucose uptake through β_3_-adrenergic receptor activation and cyclic-AMP signaling. This uptake was additive with insulin, and indicated an increased inherent activity of GLUT4 rather than its translocation. Therefore, NE can boost glucose utilization in adipose tissue, even though systemic NE excess causes whole-body insulin resistance [75].

#### 2.3.1. Sympathetic Overactivity in Early Metabolic Disease and Insulin Sensitivity

In healthy subjects, experimental norepinephrine infusion causes acute insulin resistance. Hyperinsulinemic–euglycemic clamp studies showed that a pressor dose of norepinephrine significantly reduces glucose disposal rates by increasing hepatic glucose production and decreasing peripheral glucose uptake. Earlier research confirmed that even normal elevations in NE impair glucose tolerance and insulin sensitivity [72,76,77].

An elevated sympathetic tone—assessed via increased NE spillover and augmented cardiovascular responses—is evident in obesity and prediabetes, serving as a predictor for future insulin resistance. Interventions aimed at modulating sympathetic nervous system activity, such as imidazoline receptor agonists, have shown efficacy in improving insulin sensitivity and reducing hypertension. These findings underscore the pivotal role of noradrenergic hyperactivity in the initial stages of metabolic dysregulation [71,76,78,79,80,81,82].

#### 2.3.2. Impact on Beta-Cell Function

NE inhibits insulin secretion through α_2_-adrenergic receptors on β-cells, activating G_i_/G_o_ proteins that suppress Ca^2+^ influx and exocytosis. This mechanism acutely exacerbates hyperglycemia during stress and may worsen insulin secretory capacity over time, compounding metabolic defects in individuals with prediabetes and early T2D [83,84,85].

Peripheral insulin resistance and compensatory hyperinsulinemia: Chronic sympathetic activation and NE release contribute to hepatic gluconeogenesis and skeletal muscle insulin resistance. In vivo, this resistance normally triggers compensatory β-cell hypersecretion; however, NE’s inhibitory action on β-cells can blunt this compensation [72]. Together, these data highlight that sympathetic overactivation and NE signaling contribute to both insulin resistance and β-cell dysfunction in early metabolic disease. Early detection of heightened NE tone could serve as a biomarker and a treatment target, with sympatholytic or β-blocker strategies potentially improving insulin dynamics in prediabetes and new-onset T2D [86,87,88].

### 2.4. Gamma-Aminobutyric Acid (GABA)

GABA is the primary inhibitory neurotransmitter in CNS, with emerging importance in peripheral metabolic regulation via its production by pancreatic β-cells and hepatic pathways [5,89,90,91,92].

#### 2.4.1. CNS and Hypothalamic Regulation

Within the hypothalamus, GABAergic neurons intricately regulate glucose-sensing circuits that coordinate autonomic and endocrine responses to metabolic signals. Disruptions in GABAergic inhibitory signaling may impair the hypothalamic control of energy homeostasis and blood glucose levels, potentially leading to early insulin resistance in the progression from prediabetes to T2D. Similarly, in the thalamus, GABAergic neurons play a key role in regulating the glucose-sensing circuits responsible for coordinating autonomic and endocrine responses to metabolic signals. Disruptions in GABAergic signaling can compromise these processes, possibly contributing to early insulin resistance during the development of T2D [93,94].

Within pancreatic islets, GABA is co-released with insulin from β-cells and exerts autocrine and paracrine effects via GABA_A and GABA_B receptors. It stabilizes the pulsatile pattern of insulin secretion and inhibits glucagon release from α-cells. In human islets, blockading GABA_A receptors disrupts this rhythmic secretion, underscoring GABA’s role in precise glycemic control [18,20].

Islet GABA_A receptors: Human β cells express high-affinity GABA_A receptors that respond to GABA concentrations of 100–1000 nM; activation modulates insulin granule exocytosis and enhances glucose-stimulated insulin secretion. Saturating GABA desensitizes the receptors, and β cells from donors with T2D show increased GABA affinity [95].

Animal and in vitro studies show that GABA promotes β-cell proliferation, increases insulin secretion, and decreases glucagon output. Liver-specific knockdown of GABA transaminase improves insulin sensitivity and reduces food intake in obese mice, indicating that hepatic GABA metabolism also influences systemic glucose regulation [96].

#### 2.4.2. Clinical Evidence

In a randomized placebo-controlled trial, oral GABA supplementation in individuals with prediabetes did not significantly change postprandial glucose response over 95 days. However, some secondary markers showed modest metabolic trends that warrant further investigation [97].

GABAergic pathways exert multi-level control over glucose metabolism through hypothalamic inhibitory tone, islet hormone secretory dynamics, and hepatic insulin sensitivity. Disruption in any of these systems may contribute to early metabolic dysregulation in prediabetes and new-onset T2D. While preclinical data demonstrate strong effects, human trials are limited and inconclusive, highlighting a critical need for rigorous clinical evaluation [18,25,98,99].

#### 2.4.3. Animal Evidence and Human Trials

Oral GABA treatment in high-fat diet–fed mice reduced weight gain and fasting glucose, improved glucose tolerance and insulin sensitivity, decreased adipose inflammation, and increased regulatory T-cell frequency [100].

A randomized placebo-controlled trial in prediabetic adults (*n* = 52) found that 500 mg GABA taken three times daily for 95 days did not significantly affect postprandial glucose excursions or other glycemic markers. Therefore, although GABA shows potential in cellular and animal studies, its effectiveness in human prediabetes remains uncertain [97].

### 2.5. Glutamate

Glutamate, the primary excitatory neurotransmitter in the CNS, plays a vital dual role in both central neurophysiology and peripheral metabolic regulation. In the brain, excess glutamate contributes to neuroinflammation and excitotoxicity—processes linked to diabetic neuropathy and cognitive decline in early diabetes via the overactivation of NMDA receptors and increased oxidative stress [101].

Autocrine signaling in α-cells: Human α-cells co-release glutamate with glucagon when glucose levels fall; this glutamate binds AMPA/kainate receptors on α-cells, causing depolarization, Ca^2+^ influx, and enhanced glucagon secretion. Blocking these receptors in vivo reduces glucagon release and worsens insulin-induced hypoglycemia [102].

EAAT2 glutamate transporter: Pancreatic β-cells express the excitatory amino acid transporter EAAT2 (GLT1). In islets from T2D patients, hyperglycemia downregulated PI3K/Akt signaling and caused the internalization of EAAT2, reducing glutamate uptake and increasing β-cell vulnerability. Restoring EAAT2 expression (e.g., with ceftriaxone) rescued β-cell function and survival [103].

#### 2.5.1. Central Excitatory Imbalance and Pancreatic β-Cell Glutamate Toxicity

1H-MRS studies have identified altered glutamate/glutamine ratios in individuals with early-stage T2D, suggesting excitatory pathway dysregulation before noticeable cognitive deficits [30]. Disrupted synaptic versus extrasynaptic NMDA receptor signaling may worsen neuronal oxidative stress and inflammation, impairing CNS-mediated metabolic regulation [104,105,106].

Pancreatic islets express NMDA-type glutamate receptors that, when overactivated, contribute to β-cell dysfunction under hyperglycemic stress. In rodent models, chronic high glucose levels increase islet glutamate release, which triggers NMDA receptor-mediated Ca^2+^ influx, reactive oxygen species production, and NF-κB and NLRP3 inflammasome activation, leading to impaired insulin secretion and cell death [107]. The use of NMDA receptor antagonists, such as MK-801 or memantine, protected β-cells from glucotoxicity and enhanced function both in vitro and in vivo [101,104,107].

#### 2.5.2. Systemic Biomarkers and Therapeutic Insight

Elevated plasma glutamate concentrations are consistently documented in individuals exhibiting insulin resistance and early-stage T2D, with correlations observed with oxidative stress and cardiovascular risk factors. A reduced glutamine/glutamate ratio has been suggested as a potential biomarker for metabolic stress, indicative of excitotoxic and inflammatory pathways. A literature review supports the therapeutic potential of NMDA receptor modulation in β-cells for intervention in early T2D [107,108].

Glutamatergic dysregulation acts as a dual pathogenic pathway in early metabolic disturbances, involving centrally mediated excitotoxic neuroinflammation and peripheral β-cell glutamate toxicity. These insights highlight the therapeutic potential of NMDA receptor antagonists (e.g., memantine) to help prevent early diabetogenic processes. Regulation serves as a dual pathogenic pathway in initial metabolic derangements, involving centrally mediated excitotoxic neuroinflammation and peripheral β-cell glutamate toxicity. These insights underscore the therapeutic potential of NMDA receptor antagonists (e.g., memantine) to mitigate early diabetogenic processes [107,109].

Table 1 outlines the main neurochemical features of key neurotransmitters recalled within the nervous system and their functional roles in PreDM and T2D.

## 3. Evidence of Neurotransmitter Alterations in Prediabetes

Emerging preclinical and clinical evidence indicates that neurotransmitter imbalances are detectable early in metabolic disease, offering insight into the neurochemical pathways altered before full-blown T2D manifests [1,34].

### 3.1. MRI/MRS Studies

Advanced neuroimaging techniques, particularly 1H-MRS, have consistently revealed altered levels of GABA and glutamate in key brain regions of individuals with prediabetes or early hyperglycemia. For example, a multicenter 1H-MRS study reported elevated GABA-to-glutamate ratios in the medial prefrontal cortex and thalamus of participants with prediabetes or early T2D, correlating with fasting plasma glucose and HbA1c levels. Another study demonstrated increased GABA concentrations in the medial prefrontal cortex of T2D patients, even before significant cognitive impairments occurred. These findings underline that inhibitory–excitatory neurotransmitter imbalances correlate with early metabolic markers [28,29,30,36].

### 3.2. CSF and Peripheral Neurochemical Markers

Cerebrospinal fluid (CSF) analyses provide complementary peripheral evidence. In one notable study, blood glucose levels significantly correlated with CSF concentrations of the monoamine metabolites homovanillic acid (HVA, a dopamine breakdown product) and 3-methoxy-4-hydroxyphenylglycol (MHPG, a norepinephrine metabolite) in individuals with impaired glucose tolerance. This suggests the early perturbation of central dopamine and norepinephrine turnover in prediabetes. In contrast, classical neurodegeneration markers like tau and β-amyloid showed no consistent difference in CSF between prediabetic and normoglycemic subjects, highlighting that neurotransmitter changes may occur independently of neurodegenerative pathology [28,29,117,118,119].

### 3.3. Behavioral and Cognitive Correlates

Higher GABA levels in the prefrontal cortex have been associated with lower cognitive performance scores, even at mild hyperglycemia [81,120]. This aligns with observations that altered neurotransmitter levels—particularly elevated cortical GABA—are linked to memory dysfunction in prediabetic and early T2D cohorts [119].

Additionally, non-invasive neuromodulation (e.g., transcranial magnetic stimulation), combined with MRS, has revealed dysfunctional cortical excitatory–inhibitory balance, potentially indicating early neurochemical dysregulation [121,122].

## 4. Neurotransmitter Changes in New-Onset T2D

Once T2D is clinically diagnosed, dysregulation of neurotransmitter systems becomes more pronounced, affecting central pathways, the HPA axis, and neurobehavioral outcomes [123].

### 4.1. Alterations in Neurotransmitter Levels and Receptor Expression

Research indicates that excitatory and inhibitory neurotransmitter balance deteriorates further in new-onset T2D. A recent review highlighted that increased glutamate-to-GABA imbalance exacerbates cognitive and metabolic dysfunction, mediated via disrupted NMDA receptor signaling, oxidative stress, and impaired insulin–neurotransmitter crosstalk [30,44].

Plasma metabolomic studies revealed that individuals with incident T2D exhibit persistently elevated plasma glutamate and a reduced glutamine-to-glutamate ratio—markers of excitatory overload and metabolic stress [1].

### 4.2. HPA Axis Overactivation

New-onset T2D is often accompanied by hyperactivity of the HPA axis. Elevated cortisol levels—which promote gluconeogenesis, lipolysis, and insulin resistance—are frequently reported in early T2D cases [124,125,126]. For example, patients with newly diagnosed T2D show significantly higher morning cortisol concentrations, which are inversely related to glycemic control measured by continuous glucose monitoring [127,128].

The activation of HPA axis dysfunction appears to be bidirectional: hyperglycemia itself stimulates cortisol release, while chronic cortisol exposure worsens β-cell function and insulin sensitivity [124,129,130].

### 4.3. Effects on Mood, Cognition, and Eating Behavior

Neurotransmitter alterations in early T2D not only impair metabolism, but also contribute to psychological and behavioral symptoms. Overactive glutamate-NMDA receptor signaling in the hippocampus and prefrontal cortex has been linked to neuroinflammation and mood disturbances such as anxiety and depression, which commonly occur soon after T2D onset [30,101].

Serotonergic and dopaminergic dysfunction also occurs: reductions in central serotonin and dopamine lead to altered reward sensitivity, increased emotional eating, and decreased satiety signaling—factors that sustain hyperglycemia and hinder lifestyle adherence [37,131].

Further exacerbating the cycle, chronic HPA activation suppresses hippocampal BDNF production and disrupts excitatory/inhibitory balance, negatively affecting cognition and stress resilience [132,133].

Understanding these intertwined neurochemical and endocrine alterations is crucial for devising early interventions—ranging from NMDA antagonists and cortisol modulators to psychotherapeutic strategies—that could prevent rapid disease progression and improve patient outcomes [30,37,127,132,134]:Neurotransmitter imbalance in new-onset T2D involves increased excitatory (glutamate) tone and reduced inhibitory (GABAergic) control, contributing to cognitive decline and metabolic worsening;HPA axis dysregulation, with elevated cortisol, both reflects and exacerbates early T2D pathophysiology;Neurobehavioral consequences—including anxiety, depression, and disordered eating—are linked to impaired dopaminergic and serotonergic regulation, further undermining glycemic control.

## 5. Potential Mechanisms Linking Neurotransmitter Dysfunction and Glucose Metabolism

In prediabetes and early T2D, neurotransmitter perturbations are not isolated phenomena, but are deeply entwined with core metabolic derangements. Here, we explore several central mechanistic pathways that underpin their interaction [10,135].

### 5.1. Brain Insulin Signaling and Neurotransmitter Regulation

The brain, notably the hypothalamus and hippocampus, functions as a critical insulin-responsive organ where insulin receptor signaling modulates various processes, including appetite regulation, neurotransmission, and peripheral glucose homeostasis. Binding of insulin activates the PI3K/Akt pathway, facilitates GLUT4 translocation, and influences neuronal excitability through K_ATP channels. Disruptions in this signaling cascade, indicative of central IR, perturb neurotransmitter systems such as dopamine, glutamate, and GABA, thereby contributing to maladaptive behavioral responses, synaptic dysfunction, and dysregulation of hepatic glucose production. These central abnormalities may antecede and precipitate peripheral metabolic impairments [136,137,138].

### 5.2. Inflammatory Pathways

Chronic low-grade inflammation is a hallmark of metabolic dysfunction. Elevated cytokines—IL-6, TNF-α, and IL-1β—activate JAK/STAT and NF-κB pathways, impair insulin signaling at both the insulin receptor substrate (IRS) level and in downstream cascades. They also cross the blood–brain barrier and perturb neurotransmitter systems by altering monoamine synthesis and receptor function. In rodent models, central IL-1β and IL-6 suppress hypothalamic insulin sensitivity and disrupt GABA/glutamate balance, amplifying hyperglycemia and appetite dysregulation [9,139,140,141].

Neurotransmitter perturbations and mitochondrial impairment form a vicious cycle. Hyperglycemia, excitotoxic glutamate signaling, and pro-inflammatory cytokines converge on mitochondrial pathways to generate ROS, leading to neuronal injury and synaptic dysfunction. Conversely, mitochondrial dysfunction impairs neurotransmitter synthesis, vesicular transport, and synaptic plasticity, reinforcing insulin resistance and β-cell decline. These mechanisms link central inflammation and neurotransmitter imbalance to early metabolic impairment [34,136,142].

### 5.3. Microbiota–Gut–Brain Axis and Integrated Pathophysiological Feedback

The gut microbiota and enteric nervous system (ENS) form a parallel regulatory network that influences glycemia via vagal and cytokine signaling. Dysbiosis increases circulating IL-6 and TNF-α, activates ENS inflammation, and disrupts vagal neurotransmitter signals (e.g., GABA and serotonin) to the hypothalamus. This feedback loop potentiates central insulin resistance, HPA axis activation, and metabolic dysfunction [143,144,145].

Central IR, inflammation, oxidative stress, and gut–brain dysbiosis constitute an interconnected network of pathophysiological mechanisms that perturb neurotransmitter systems. This dysregulation synergistically impairs both central and peripheral glucose homeostasis. In early-stage T2D, targeted interventions at these nodes—using insulin sensitizers, anti-inflammatory agents, antioxidants, or microbiome modulation—may facilitate the restoration of neurochemical equilibrium and metabolic function [146,147,148].

Some of these pathways interact in a self-reinforcing way, as shown in Table 2.

## 6. Future Directions and Gaps

Despite growing evidence linking neurotransmitter dysfunction to early metabolic disease, significant gaps remain in our understanding of how these neurochemical changes develop and influence prediabetes and T2D progression.

### 6.1. Longitudinal Human Studies

Most current data derive from cross-sectional or animal studies, limiting causal inference. Longitudinal studies that track neurotransmitter changes—via MRS, CSF biomarkers, or plasma metabolomics—are urgently needed to determine whether these neurochemical alterations precede, accompany, or follow glycemic deterioration [1,29].

Sex-specific neuroendocrine regulation remains underexplored. Estrogen and testosterone modulate neurotransmitter systems differently, influencing both central insulin sensitivity and feeding behavior. Yet, few studies stratify by sex, limiting the development of tailored interventions [153,154].

There is a pressing need to develop predictive models that incorporate neurotransmitter profiles alongside traditional metabolic markers. For example, integrating glutamate/GABA ratios or dopamine turnover metrics with clinical indicators (HbA1c, HOMA-IR, BMI) could improve early risk stratification for T2D. Machine learning applied to neuroimaging and biochemical data may further refine these personalized predictions [11].

### 6.2. Multi-Omics and Neuroimaging Integration

Future research should adopt systems biology approaches combining genomics, metabolomics, proteomics, and neuroimaging. For example, combining 1H-MRS–derived neurotransmitter levels with transcriptomic profiles from the hypothalamus or islets could unravel novel neuro-metabolic pathways. The Human Connectome Project and UK Biobank offer potential platforms for such integrative analyses [1].

To fully understand the role of neurotransmitter dysfunction in diabetes pathophysiology, future studies must
Employ longitudinal, sex-aware, and mechanistic designs;Integrate neuroimaging with multi-omics data;Build toward precision diagnostics and targeted neuro-metabolic therapies.

Below, Figure 2 illustrates a mind map of all of the interactions of neurotransmitters in prediabetes and T2D.

## 7. Conclusions

Neurotransmitter systems—including dopamine, 5-HT, NE, GABA, and glutamate—are central to energy balance, appetite, glucose metabolism, and neurobehavioral regulation. Dysregulation of these circuits is evident not only in established T2D, but also during prediabetic stages, preceding overt hyperglycemia and pancreatic dysfunction. Mechanistically, neurotransmitter dysfunction acts both as a consequence and a driver of insulin resistance, β-cell stress, hypothalamic inflammation, and HPA axis activation. This neuro-metabolic crosstalk forms a feedback loop in which central insulin resistance disrupts neurotransmitter signaling, exacerbating glycemic imbalance, appetite dyscontrol, and cognitive decline.

These insights highlight neurotransmitter imbalance as a key, dynamic factor in the development of diabetes, rather than a secondary complication. Therefore, neurobiological markers should be included in risk assessment and early intervention strategies. Non-invasive imaging, peripheral biomarkers (such as glutamate/glutamine ratio), and therapies that modulate neurotransmitters show promise for identifying high-risk individuals and guiding personalized metabolic treatment. Ultimately, targeting neurochemical pathways could allow for the earlier diagnosis and intervention of T2D, especially during its initial reversible stages, which can help prevent permanent organ damage.

## Figures and Tables

**Figure 1 ijms-26-07847-f001:**
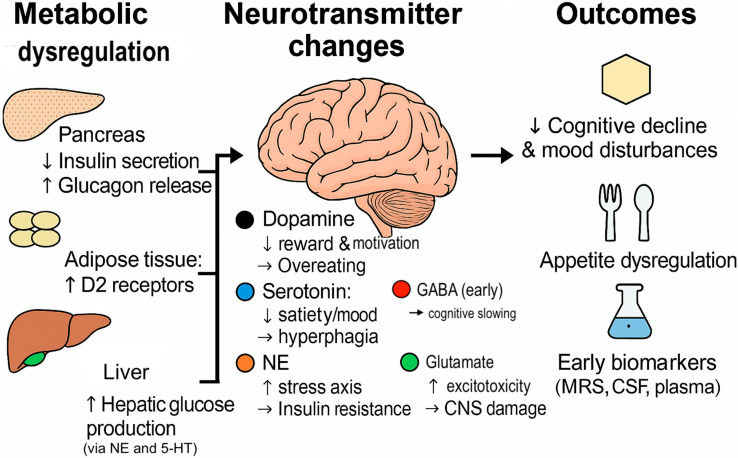
Neurochemical imbalance in early diabetes: a dual CNS–metabolic axis. The figure illustrates metabolic dysregulation in the pancreas (↓ insulin secretion, ↑ glucagon release), adipose tissue (↑ D2 receptors), and liver (↑ hepatic glucose production via norepinephrine (NE) and serotonin (5-HT)); central neurotransmitter changes (↓ dopamine and serotonin, ↑ NE, early ↑ GABA, ↑ glutamate); and outcomes such as cognitive decline, mood disturbances, appetite dysregulation, and early biomarkers. NE = norepinephrine; 5-HT = serotonin; MRS = magnetic resonance spectroscopy; CSF = cerebrospinal fluid; ↑ = activate; ↓ = suppress (Figure created using BioRender).

**Figure 2 ijms-26-07847-f002:**
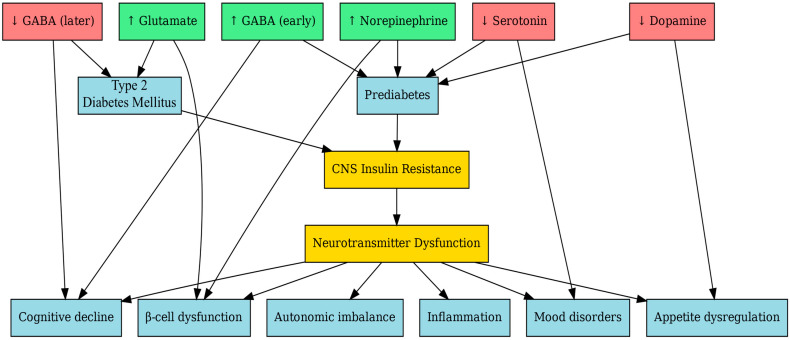
Mind map of neurotransmitter dysfunction in prediabetes and T2D. Decreases in neurotransmitters are shown in red, increases in green, and the dysfunction node in yellow to help differentiate categories. This mind map summarizes the key neurotransmitter alterations in prediabetes and new-onset T2D. In prediabetes, early-stage imbalances include reduced dopamine and serotonin and elevated GABA, glutamate, and norepinephrine—affecting reward processing, satiety, and cognition. Notably, GABA levels may initially rise as a compensatory mechanism, potentially contributing to cognitive slowing. As the disease progresses to T2D, neurotransmitter dysregulation intensifies: glutamate becomes excitotoxic, GABA levels drop (reflecting impaired inhibitory tone both centrally and in pancreatic islets), and cortisol increases due to HPA axis overactivation. These changes are compounded by systemic inflammation and contribute to cognitive decline, mood disorders, appetite dysregulation, and autonomic imbalance (Figure created using BioRender).

**Table 1 ijms-26-07847-t001:** Summary of Neurotransmitters implicated in PreDM and T2D.

Neurotransmitter	Source	Receptors	Effect on Insulin Secretion	Effect on Glucagon Secretion	Impact on β-Cell Health	CNS/Peripheral Mechanisms	Therapeutic Implications
**Dopamine (DA)** [12,13,40]	β-cells, CNS, adipose tissue	D2, D1/D2 heteromers	↓ via D2R activation	↑ via central mechanisms	↓ induces apoptosis under stress	CNS reward disruption; adipose D2R upregulation	Bromocriptine, cabergoline, D2 agonists
**Serotonin** **(5-HT)** [52,110,111,112,113]	Enterochromaffin cells, β-cells	5-HT1F, 2B, 3R	↑ via serotonylation and 5-HT2B/3R	↓ via 5-HT1F	↑ proliferation, protects against metabolic stress	Regulates appetite, mood, satiety via 5-HT2C (CNS)	SSRIs (sertraline); microbiota modulation
**Norepinephrine** **(NE)** [83,114]	Locus coeruleus, SNS terminals	α2-adrenergic	↓ via α2R on β-cells	↑ via SNS tone	↑ glucotoxicity stress; ↓ insulin signaling	SNS overactivity worsens IR, promotes HPA activation	Imidazoline receptor agonists, β-blockers
**GABA** [18,20,115]	β-cells, CNS, gut microbiota	GABA A, GABA B	↑ pulsatile secretion; stabilizes release	↓ via paracrine inhibition	↑ proliferation, anti-inflammatory	Hypothalamic glucose sensing; hepatic GABA metabolism	GABA supplementation, vagal stimulation
**Glutamate** [101,116]	α-cells, CNS	NMDA, AMPA/ kainate	↓ via NMDA excitotoxicity	↑ via AMPA/kainate on α-cells	↓ induces β-cell death	CNS excitotoxicity, ROS generation, neuroinflammation	NMDA antagonists (memantine)

↑ = Stimulates or enhances; ↓ = inhibits or reduces; CNS = Central Nervous System; HPA axis = Hypothalamic–Pituitary–Adrenal axis; NMDA = N-methyl-D-aspartate; AMPA = α-amino-3-hydroxy-5-methyl-4-isoxazolepropionate.

**Table 2 ijms-26-07847-t002:** Neuroendocrine mechanisms linking neurotransmitter dysregulation to metabolic dysfunction in early T2D [149,150,151,152].

Mechanism	Effect on Neurotransmitters	Consequent Metabolic Impact
Insulin resistance (central)	Dysregulated dopamine, glutamate, GABA	Increased appetite, hepatic glucose output
Inflammation	Monoamine/peptide imbalances	Impaired central regulation, peripheral IR
ROS and mitochondrial stress	Impaired synaptic function	Cognitive decline, further IR
Gut–brain signals	Disrupted vagal neurotransmission	HPA activation, glycemic dysregulation

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
