# Peer review of "Neurotransmitter Alterations in Prediabetes and Type 2 Diabetes Mellitus: A Narrative Review"

_ijms, 2025, doi:10.3390/ijms26167847_

Round 1

Reviewer 1 Report

Comments and Suggestions for Authors

Good timely topic to discuss the relationship between brain neurotransmission dysfunction and diabetes. As of now, the manuscript lacks enough discussions on the findings between each mentioned neurotransmitter and diabetes and/or related metabolic dysfunctions. The current descriptions and discussions are only at the surface. A substantially more detailed discussion (animal, human studies) is needed that can greatly improve the manuscript to the publication form.

There are many citations of review papers which is not appropriate or informative for a review manuscript when it comes to discussing the roles and findings of neurotransmitters in glucose homeostasis, ingestive behavior, and insulin sensitivity. Authors need to refer to the original research articles a lot more.

Author Response

We sincerely thank the reviewer for the thoughtful and constructive comments. We have carefully considered all the concerns raised and have made substantial revisions to strengthen our manuscript. In particular, we expanded the discussion on the relationship between each neurotransmitter and diabetes/metabolic dysfunctions, incorporating both animal and human studies to provide deeper insights beyond surface-level descriptions.

We also acknowledge the reviewer’s valuable suggestion regarding citations. In the revised version, we have substantially increased references to original research articles and minimized reliance on review papers, ensuring that our discussions are better supported by primary evidence.

We truly appreciate the reviewer’s guidance, which has helped us to significantly improve the manuscript. We remain open to further constructive criticism to continue refining our work toward its best potential.

Reviewer 2 Report

Comments and Suggestions for Authors

In present work, AhriÈ›culesei et al. try to review the alterations in key neurotransmitter systems—dopamine, serotonin, norepinephrine, gamma-aminobutyric acid, and glutamate—in individuals with prediabetes and newly diagnosed type 2 diabetes. This paper indicates that neurotransmitter imbalances should be viewed as early, dynamic contributors to development of diabetes complications. In addition, non-invasive imaging, peripheral biomarkers (e.g., glutamate/glutamine ratio), and neurotransmitter modulating therapies may help identify high-risk individuals and personalize metabolic care. However, there are some questions that should be explained.

Major concerns

  1. Figure 1 should be revised. The pancreas, adipose and liver should be corresponding to their functions one by one. The biomarkers are in plasma? The images for blood should be revised. In addition, the image for appetite dysregulation also should be revised. Furthermore, some abbreviations should be explained.
  2. Figure 2 should be revised. The words are small, and different colour should be used.
  3. English grammar and writing should be checked and revised throughout the manuscript.

Minor concerns

  1. Line 22, delete ‘(GABA)’.
  2. Line 24, delete ‘(CNS)’.
  3. ‘Type 2 diabetes mellitus (T2D)’ or ‘type 2 diabetes (T2D)’? There are many ‘type 2 diabetes (T2D)’ and ‘type 2 diabetes’ throughout this manuscript, which should be revised to ‘T2D’.
  4. Line 140, please explain ‘D2 receptor’. ‘2’ is in subscriptor not.
  5. Line 145, please explain ‘D2 signaling’.
  6. Line 158, delete ‘(Tph1)’.
  7. Lines 180-182, short paragraph. Please revise, and check this throughout this manuscript. There are many short paragraphs in this article.
  8. Line 227, ‘Gamma-Aminobutyric Acid (GABA)’ should be italic or not. Please check this throughout this manuscript for subsection.
  9. Line 282, correct ‘mechanisms . Review’.
  10. Line 290, correct ‘how they acts in’.
  11. Lines 318 and 331, revise ‘&’ to ‘and’. Please check this throughout this manuscript.
  12. Line 384, correct ‘regulation . Insulin’.
  13. Tables 1 and 2, reference numbers should be added in the table.
  14. Conclusions section, there are so many paragraphs, which should be revised.
  15. The reference format is not consistent. Journal name of some references are in abbreviation, but other references are not. Please check these throughout Reference section, and delete ‘Volume’ in some references.
Comments on the Quality of English Language

The English could be improved to more clearly express the research.

Author Response

We would like to sincerely thank the reviewer for the detailed and constructive feedback provided on our manuscript. We carefully addressed both the major and minor concerns, as we are committed to improving the clarity, precision, and overall quality of our work.

Regarding the figures, we have revised Figure 1 to better align the pancreas, adipose tissue, and liver with their corresponding functions, clarified the localization of biomarkers, and updated the illustrations for blood and appetite dysregulation. We also revised Figure 2 by adjusting the font size and improving the color scheme for readability.

We thoroughly revised the manuscript for English grammar and writing style, ensuring consistency and readability throughout. In addition, we carefully addressed all minor comments: unnecessary abbreviations were removed, terms such as D2 receptor and D2 signaling were explained, formatting inconsistencies were corrected, and short paragraphs were revised for better flow. Tables 1 and 2 now include reference numbers, and the Conclusion section was restructured for conciseness and clarity. Furthermore, we standardized the reference formatting, ensuring consistency across journal names, abbreviations, and citation style.

We are deeply grateful for these insightful comments, which helped us significantly refine and improve our manuscript. We remain fully open to further constructive criticism that will allow us to reach the best possible version of this review.

Reviewer 3 Report

Comments and Suggestions for Authors

In the manuscript “Neurotransmitter Alterations in Prediabetes and New-Onset Type 2 Diabetes: A Narrative Review”, the authors explore the mechanistic links between neurotransmitter dysfunction and glucose metabolism.

There are several issues and observations within the manuscript that warrant further evaluation.

  1. References should be cited after each relevant sentence, rather than grouped at the end of the paragraph, particularly when the information presented is not uniformly supported by all the cited sources.
  2. The in-text citations should be carefully reviewed for accuracy. For example, the statement on lines 121–123 (“For instance, elevated GABA concentrations in the mPFC have been associated with insulin resistance and cognitive impairment in T2D patients”) appears to be based on findings from reference 23, yet this source is not cited here.
  3. Lines 186–188: The statement about the effects of SSRIs on food intake is too broad. The impact varies by compound. For example, fluoxetine is approved for the treatment of bulimia and may reduce food intake. This variability should be acknowledged to provide a more accurate information regarding SSRIs effects.
  4. In Subchapter 2.3.3 Impact on Beta-cell Function, the presentation of information lacks clarity. The authors state that norepinephrine (NA) reduces insulin secretion but then appear to link this with increased peripheral insulin resistance, a condition that typically leads to increased insulin secretion and compensatory β-cell mass expansion. This apparent contradiction should be clarified. Furthermore, the suggestion that beta-blockers could improve insulin dynamics requires additional explanation. Specifically, the authors should elaborate that beta-blockers may exert this effect by reducing sympathetic tone through central mechanisms.
  5. The quality of the images is suboptimal and should be improved to enhance readability and visual clarity.

Optionally, the authors may consider including a brief discussion of central leptin resistance in the Introduction. Given its relevance to energy balance, appetite regulation, and neuroendocrine signaling, addressing leptin resistance could help contextualize the broader neurochemical alterations observed in prediabetes and early T2D.

Author Response

We are very grateful to the reviewer for the careful reading of our manuscript and the constructive suggestions, which have been extremely valuable in improving both the accuracy and clarity of our review.

We addressed the concern regarding references, for instance, the statement on lines 121–123 regarding GABA concentrations in the mPFC has been corrected to cite the appropriate reference (previously missing).

We also revised the section on SSRIs and food intake to reflect compound-specific differences. In particular, we highlighted fluoxetine’s unique role in reducing food intake, while acknowledging variability across other SSRIs, thus providing a more nuanced and accurate discussion.

The subsection 2.3.3 Impact on Beta-cell Function has been thoroughly revised to remove the ambiguity. We clarified the distinction between norepinephrine’s inhibitory effects on insulin secretion and the compensatory mechanisms observed in insulin resistance. Additionally, we expanded the explanation regarding beta-blockers, noting that their potential benefits may be mediated through reducing sympathetic tone via central mechanisms.

To improve the overall presentation, we enhanced the quality and clarity of all figures, ensuring readability and visual consistency.

Finally, following the reviewer’s optional but highly valuable suggestion, we introduced a brief discussion on central leptin resistance in the Introduction to better contextualize the interplay between neurochemical alterations, appetite regulation, and energy balance in prediabetes and early T2D.

We sincerely appreciate these constructive comments, which have significantly strengthened the manuscript. We remain open to further feedback that can help refine our work toward its best possible version.

Round 2

Reviewer 1 Report

Comments and Suggestions for Authors

Authors properly addressed the reviewer's comment on describing the findings from other studies and citing the original studies. 

While improved, the manuscript appears unnecessarily segmented into various subtopics without much (somewhat detailed) explanation of relevant studies and "discussions" on the association between different neurotransmitters and metabolic dysfunctions. Filling in some important study details for each subheadings (if still decide to include) would be necessary, otherwise this is not considered an in-depth review.

Author Response

We are deeply grateful to the reviewer for their careful assessment and invaluable insights regarding our manuscript. The constructive feedback we have received is of tremendous importance, as it highlights both the strengths and the areas where our work can be refined to achieve greater clarity, precision, and scientific rigor.

We view this review process as a collaborative effort in improving the manuscript, and we are very eager to act upon your recommendations. Your thoughtful suggestions will significantly enhance the manuscript’s structure and impact, and we are committed to making thorough revisions to address them in detail. We sincerely appreciate the reviewer’s expertise, time, and dedication, which we regard as essential contributions to helping us present our work at its highest standard. We remain enthusiastic and willing to continue refining the manuscript further to ensure that it fully reflects the excellence encouraged by your constructive critique.

Comment:

  • While improved, the manuscript appears unnecessarily segmented into various subtopics without much (somewhat detailed) explanation of relevant studies and "discussions" on the association between different neurotransmitters and metabolic dysfunctions. Filling in some important study details for each subheadings (if still decide to include) would be necessary, otherwise this is not considered an in-depth review.

Response: 

We have carefully refined the structure of the manuscript by reviewing the subheadings and removing or combining those that were unnecessary or redundant. This was done with the aim of improving clarity, readability, and logical flow. At the same time, we remain very open to the reviewer’s perspective: if it is felt that additional information is needed in any subheadings and it would strengthen the manuscript, or that certain topics would benefit from being expanded into their own sections, we would be more than willing to incorporate these adjustments and further develop the discussion as suggested.

Reviewer 2 Report

Comments and Suggestions for Authors

Thanks for author’s responses. However, there are STILL some questions that should be explained.

  1. Figure 1 should be revised. Some cartoon images should be included as version 1.
  2. Figure 2 should be revised. Some arrows are not displayed.
  3. Too many paragraphs for Conclusions section. Refine it.
  4. The reference format is STILL not consistent. Journal name of some references are in abbreviation, but other references are not. Please check these throughout Reference section. For example, Ref. 36, ‘Hum Brain Mapp’, or Ref. 37, ‘Human Brain Mapping’.
Comments on the Quality of English Language

The English could be improved to more clearly express the research.

Author Response

We would like to express our deepest gratitude to the reviewer for their thorough evaluation and insightful recommendations. We are genuinely honored by the time, care, and expertise invested in critically examining our manuscript. Your feedback not only underscores important areas of refinement but also offers us a valuable opportunity to strengthen the quality, clarity, and overall impact of our work.

We are truly committed to implementing your suggestions with the utmost attention and care, as we firmly believe that your perspective is helping us elevate this manuscript to its fullest potential. We are highly motivated to continue revising and improving the paper, guided by your constructive comments, so that it may serve the scientific community at the highest possible standard. We sincerely appreciate your contribution to this process and remain open and eager to make any additional changes necessary to ensure that the final version reflects the excellence that your review has encouraged us to pursue.

Comment 1 and 2: 

  • Figure 1 should be revised. Some cartoon images should be included as version 1.
  • Figure 2 should be revised. Some arrows are not displayed.

Response 1 and 2: With respect to the two figures, we have carefully revised and refined them to improve their clarity, resolution, and overall quality so that they more accurately illustrate the key concepts of our review. We have made every effort to ensure that the figures meet the highest standards for readability and scientific accuracy. Should the reviewer feel that additional adjustments would further enhance their presentation or interpretability, we would be more than happy to implement such suggestions.

Comment 3: 

  • Too many paragraphs for Conclusions section. Refine it.

Response 3: As for the Conclusions section, we have carefully revised and trimmed it in order to make it more concise, focused, and aligned with the journal’s style. We hope that the revised version now provides a sharper summary of our findings and their implications. If the reviewer feels that further adjustments could improve its balance or impact, we would be very glad to address those recommendations.

Comment 4:

  • The reference format is STILL not consistent. Journal name of some references are in abbreviation, but other references are not. Please check these throughout Reference section. For example, Ref. 36, ‘Hum Brain Mapp’, or Ref. 37, ‘Human Brain Mapping’.

Response 4: We have now carefully refined the reference list and replaced abbreviations where appropriate to ensure consistency with the journal’s guidelines. Every effort has been made to double‑check accuracy and formatting. However, if any mistakes remain, we are fully open and willing to correct them immediately in accordance with the  recommendations.

Round 3

Reviewer 1 Report

Comments and Suggestions for Authors

Well addressed. One thing unclear is the way subheadings are written or combined as a separate titles or sentences. This has to be fixed throughout the manuscript.

Example: 2.4.1. CNS and Hypothalamic Regulation. Pancreatic Islet Influence. Peripheral Metabolic Effects

Author Response

Thank you very much for your thoughtful and constructive opinion. Your feedback has been extremely valuable in strengthening the work, and we greatly appreciate the time and expertise you invested in the process.

Comment:

  • Well addressed. One thing unclear is the way subheadings are written or combined as a separate titles or sentences. This has to be fixed throughout the manuscript. Example: 2.4.1. CNS and Hypothalamic Regulation. Pancreatic Islet Influence. Peripheral Metabolic Effects

Response: Did it.